# Extracellular Vesicles as Targeted Communicators in Complementary Medical Treatments

**DOI:** 10.3390/ijms26125896

**Published:** 2025-06-19

**Authors:** Keehyun Earm, Yung E. Earm, Denis Noble

**Affiliations:** 1Moon Soul Graduate School of Future Strategy, Korea Advanced Institute of Science and Technology, 291 Daehak-ro, Yuseong-gu, Daejeon 34141, Republic of Korea; kh.earm@kaist.ac.kr; 2Department of Physiology, College of Medicine, Seoul National University, Seoul 03080, Republic of Korea; earmye@snu.ac.kr; 3Biomedical Science & Engineering Major in Interdisciplinary Studies, Daegu Gyeongbuk Institute of Science and Technology, Daegu 42988, Republic of Korea; 4Department of Physiology, Anatomy & Genetics, University of Oxford, Oxford OX1 3PT, UK

**Keywords:** exosomes, extracellular vesicles, meridians, pangenesis, BongHan System

## Abstract

The supposed meridians of traditional oriental medicine have been a cause of conflict between traditional and modern medical science. A possible resolution has been proposed: That extracellular vesicles, including exosomes, may be the transmitters of traditional therapies such as massage and acupuncture. This article develops that idea by proposing that the pathways between surface and deep structures may be laid down during the embryonic migration of cells from one region of the developing body to distant regions. This hypothesis depends on the proven targeting of vesicular communication via cell surface binding molecules and their complementary binding sites on target cells. The hypothesis is therefore experimentally testable. The article also draws attention to a strong analogy with Charles Darwin’s theory of pangenesis for particulate communication between the soma and germline.

## 1. Introduction

Most forms of traditional medicine in East Asia derive from ideas of the anatomy and physiology of the body in classical Chinese medicine, developed at least as early as the second century BCE [1], but possibly much earlier, as indicated by excavations such as the *Mawangdui* tomb excavation, dating from 168 BCE [2]. A central feature of this representation of the body is that communication pathways exist between specific areas of the surface of the body and corresponding internal parts of the body, including internal organs. Traditionally, these pathways are called meridians, an idea that has formed a major stumbling block to any integration of modern Western and traditional Eastern medical traditions (see, e.g., the Wikipedia entry on Traditional Chinese Medicine, https://en.wikipedia.org/wiki/Traditional_Chinese_medicine (accessed on 13 May 2025)). Yet, evidence for the existence of the meridians is necessary for any Western-style scientific understanding of how treatments like acupuncture and massage may work causally. But Western anatomical and physiological analysis failed to find channels that could explain such specific effects transmitted from surface stimulation to specific internal organs.

One of the last attempts to perform so within the framework of macro-anatomy was made by a Korean scientist (see review by Liu et al. [3]), known as the BongHan System [4,5], speculating that a third circulatory system exists in addition to the arteries and veins [6]. But, intriguing though the BongHan System is, it does not explain the specificity of communication: Why should stimulation of a particular part of the body surface specifically affect particular tissues and organs far away from that area?

## 2. Fields and Hidden Micro-Level Processes

A standard way in which Western science has sought to resolve this kind of issue is to suppose that hidden variables or fields of influence must exist. The invention of the idea of a magnetic field is an apposite example. Postulating that a field exists then enables action at a distance to become possible. This example led, in due course, to the development of the theory of electromagnetism, without which many important discoveries in physics could not be understood.

Although it has not been widely accepted by modern evolutionary biologists, there was also a similar development in biological science during the 19th century. The originator of the theory of Natural Selection, Charles Darwin, was convinced that, on its own, that theory could not explain the origin of species by a process of evolution. One of his reasons was that, even in his book *The Origin of Species* [7], he outlined several examples that, in his view, required yet another form of action at a distance. He meant the acquisition of acquired characteristics by the progeny, an idea that was universally accepted in 18th/19th-century biology, including notably by Lamarck (1809) [8,9]. This idea led Darwin to propose that there must be communication between the soma and the germline in order for those characteristics acquired by the soma to pass to the future eggs and sperm. Darwin was therefore faced with a similar dilemma: Where were the relevant communication pathways?

He was therefore compelled to postulate action at a distance between the soma and the germline. In his 1868 book, *The Variation of Animals and Plants under Domestication* [10], he outlined his theory of pangenesis, according to which tiny particles, which he called gemmules, could communicate between soma cells and germline cells. He openly admitted there was, at that time, no experimental evidence for their existence. After Francis Galton (1871, see history in Liu 2008 [11]) performed blood transfusion experiments that failed to show the expected effects, Darwin’s pangenesis hypothesis was dropped.

## 3. Vesicles and Exosomes as Darwin’s Gemmules

Darwin’s problem was that 19th-century light microscopy did not permit people to visualize what might correspond to his gemmules. That is one reason why his pangenetic idea was unsupported by later evolutionary biologists, so his idea was ignored. August Weismann sealed the fate of the pangenesis theory with the invention of the Weismann Barrier idea in 1883 [12].

The introduction of powerful electron microscopy in the 1940s transformed this situation by showing that many particles small enough to be beyond the resolution of the light microscope are found around cells. When first discovered around 30 years ago [13], it was thought that this represented cell debris. In itself, this is not an implausible idea. The autophagy process in yeast, for example, consists of cell debris (degraded proteins and other non-functional macromolecules) being accumulated in a cell vacuole that eventually ferries the damaged molecules across the cell membrane to be recycled by the yeast colony [14].

Using fluorescent microscopy to identify RNA, proteins, and other molecules in the vesicular material led to a reinterpretation of cell ‘debris’. Extracellular vesicles have now been shown to communicate information from one population of cells in the body to cells in the same tissue but also to other populations, including notably from the soma cells to the germline cells [15,16,17,18]. This communication between the soma and the germline is similar to what Charles Darwin postulated in 1868 as his theory of pangenesis, which has now been vindicated by modern techniques identifying vesicles that can transmit control RNAs and transcription factors affecting expression of DNA in the germ cells [18].

Table 1 shows the historical timeline of the developments described in this article. 

## 4. Vesicular Theory as the Basis of Meridians

This article develops a theory based on the idea that transmission via EVs and exosomes may also form the pathways of communication often called meridians in Chinese, Korean, Japanese, and other East and South Asian medical treatments, including massage and acupuncture. Such a theory is needed, since no other pathways have been identified anatomically that could correspond to the meridians. The theory that extracellular vesicles may form such pathways is based on the fact that mechanical, electrical, and heat stimuli readily cause cells to release EVs targeted at specific organs of the body. The theory is also open to experimental tests. If the theory is correct, the postulated meridians could correspond to communication between cell populations that were close together in the embryo continuing to communicate when they are far apart in the adult. This would explain why stimuli to particular surface body regions can benefit the internal organs of the body. 

We are not the first to highlight that extracellular vesicles may be the medium for the meridians. Several recent articles in journals of traditional, alternative, and complementary medicine [20] and in Western medical journals [21,22] have already done so. Those articles, and our own, reflect a topic of discussion that is frequently arising in conferences on traditional medicine. The reason is clear. The discovery of exosomes and EVs and their targeted functional roles has completely transformed discussion of the meridians. It is the clear function of these vesicle messengers from all cells in the body to communicate with other cells in the body. That already satisfies part of the presumed role of the meridians in traditional medicine, as we will show in a later section (see Section 6). As Lyu et al. conclude, “The emergence of exosomes seems to provide a feasible way to fully simulate the therapeutic effects of acupuncture, assuming that exosomes can become not only the transmitters during acupuncture treatment but also carriers of needle-like effects at the end of an acupuncture intervention [22]. Therefore, exosomes are very interesting and promising research components.”

However, this fact in itself does not explain why particular regions of the *surface* of the body communicate *specifically* with particular *internal* parts of the body, since it does not explain *how* those particular connections arose in the first place. The specificity exists, but how did it get established during development?

## 5. Proposed Theory for Development of Meridians

We propose that ‘development’ may be the key to an answer to this question. No internal organs of the body exist at the gastrulation stage of embryonic development. Those organs develop as the result of cellular migration from parts of the early ectoderm, mesoderm, and endoderm, the patterns of which are only partially clarified so far. The reason is that tracking migrating cells during development, or indeed during any other process, such as cancer metastasis, is extremely time-consuming using existing methods. Automated tracking methods are being introduced into developmental studies [23], and it is to be hoped that such methods may enable the hypothesis we propose to be tested in future work.

Our hypothesis can be summarized as follows:Vesicular communication is a key feature of multicellular development since it provides a process by which cells in the same tissue or region identify with and conform to the regulatory processes that must exist in multicellular organisms to enable them to develop. Those communication processes develop in the embryo.It is then plausible that cells that developed such communication early in embryonic development may still possess those communication connections when they become far apart in embryonic development.There is then no difficulty in understanding how surface therapeutic interventions may influence internal organs and systems. It would be a perfectly natural outcome of embryological development.

Our proposal also leads to a clear prediction: When we know more about cell migration in embryonic development, the hypothesis predicts that the meridians may simply be the persistence of those communication memories. Thus, the liver originates from cells in the endoderm layer, one of the three germ layers formed early during gastrulation [24]. So also do the thyroid and the pancreas. The nervous system develops from cells in the ectoderm, which form the neural tube and neural crest. Germ cells also have specific origins in the early embryo before the cells migrate to form the adult germ cells.

## 6. Molecular Composition and Targeting Mechanisms of Vesicular Communication

In contrast to the situation when EVs were first discovered using electron microscopy and were regarded as cell debris, it is now well-established that many EVs are targeted at particular locations and functions. One of the authors of this article (DN) was a co-editor of a Clinical Compendium on exosomes [13], which includes articles across a wide range of clinical conditions in which vesicles have been shown to have functional targeting, ranging through various forms of cancer [25,26,27], bacterial infections [28], HIV-1 infection [29], parasitic diseases [30], cardiovascular diseases [31], dermatological disease [32], nephrology [33], neurodegenerative disorders [34], fibrosis [35], inflammation [36], metabolic syndromes [37], reproductive medicine [38], respiratory diseases [39], retinal disease [40], and regenerative medicine [41], in addition to basic biology [17,41,42].

In this section, we have chosen to highlight the following articles for their novel insights.

He et al. [43] showed exosomal targeting and its potential clinical application (Table 2). Under the action of a content sorting mechanism, some specific surface molecules can be expressed on the surface of exosomes, such as tetraspanin protein and integrin. To some extent, these specific surface molecules can fuse with specific cells so that exosomes show specific cell natural targeting. Despite the promise of exosomes as drug delivery vehicles, challenges remain in harnessing their natural targeting capabilities. Optimizing exosomal targeting is essential for improving delivery specificity and developing more effective therapeutic strategies. He et al. highlights the inherent targeting ability of exosomes and summarizes current engineering approaches—such as surface modification with targeting peptides or proteins and physical or chemical alterations—designed to enhance targeting efficiency [43]. These advancements offer new directions for disease-specific treatments and broaden the potential for clinical translation.

Hoshino et al. showed that tumor exosome integrins determine organotropic metastasis and demonstrated that integrins on exosomes dictate their organotropic metastasis patterns [44]. Exosomes from mouse and human lung-, liver-, and brain-tropic tumor cells fuse preferentially with resident cells at their predicted destination, namely lung fibroblasts and epithelial cells, liver Kupffer cells, and brain endothelial cells. Tumor-derived exosomes taken up by organ-specific cells prepare the pre-metastatic niche. Treatment with exosomes from lung-tropic models redirected the metastasis of bone-tropic tumor cells.

**Table 2 ijms-26-05896-t002:** Engineering exosomes as drug delivery systems (from He et al. [43]).

Targeting Peptide/Protein	Receptor	Target Cells/Organ	Function	Reference
IRGD peptide	Lamp2b	Breast cancer cell	Targeting delivery of DOX and effectively inhibit tumor growth	[45]
CSTSMLKAC peptide	Lamp2b	Ischemic myocardium	Reduce inflammation, apoptosis and fibrosis, enhance angiogenesis, and cardiac function	[46]
c (RgdyK) peptide	Integrin ovß3	Ischemic brain injury area	Targeting delivery of curcumin inhibits the inflammatory response in lesion area	[45]
RGE peptide	Neurokinin-1	Glioma	Targeting delivery of cur	[47]
c-Met binding peptide	c-Met	TNBC cells	Targeting delivery of DOX	[48]
GEII peptide	EGFR	Breast cancer cell	Targeting delivery of the tumor inhibitory miRNA	[49]
RVG peptide	Lamp2b	Brain neurons, microglia, and oligodendrocytes	Targeting delivery of siRNA and knockdown of Alzheimer’s disease-related genes	[50]
RVG peptide	Acetylcholine receptor	Neuron cell	Targeting delivery opioid receptor mu siRNA to treat morphine addiction	[51]
ApoA1	SR-Bl receptor	Liver cancer cells	Targeting delivery Functional miR-26a	[52]

Acronyms: DOX is Doxorubicin; IRGD peptide is a 9 amino acid cyclic peptide; CSTSMLKAC peptide is a 9 amino acid cyclic peptide that mimics endogenous peptide sequences; RGE peptide facilitates cell adhesion; GEII peptide is a peptide used as a colloidal drug delivery system to deliver anti-cancer drugs; RVG peptide is derived from Rabies virus glycoprotein; ApoA1 is a lipoprotein whose function is to transport lipids.

Alvarez-Erviti et al. showed delivery of siRNA to the mouse brain by systemic injection of targeted exosomes [50].

El-Andaloussi et al. showed exosome-mediated delivery of siRNA in vitro and in vivo [53]. The use of small interfering RNAs (siRNAs) to induce gene silencing opens a new avenue in drug discovery. However, their therapeutic potential is hampered by inadequate tissue-specific delivery. Exosomes are promising tools for drug delivery across different biological barriers. They showed how exosomes derived from cultured cells can be harnessed for the delivery of siRNA in vitro and in vivo.

Urabe et al. showed that extracellular vesicles are involved in the development of organ-specific metastasis [54]. Extracellular vesicles are increasingly being demonstrated as critical mediators of bi-directional tumor-host cell interactions, controlling organ-specific infiltration, adaptation, and colonization at the secondary site. EVs govern organotropic metastasis by modulating the pre-metastatic microenvironment through upregulation of pro-inflammatory gene expression and immunosuppressive cytokine secretion, induction of phenotype-specific differentiation, and recruitment of specific stromal cell types.

As described by Frolova and Li [55], extracellular vesicles can be functionalized with surface molecules to improve their targeting capabilities. This can be achieved either by directly attaching targeting ligands to the EV surface or by engineering the parent cells to express specific targeting moieties (see Figure 1).

## 7. Therapeutic Potential and Use as Markers of Disease States

Extracellular vesicles have emerged as pivotal players in intercellular communication, participating in both physiological and pathological processes. Due to their capacity to transfer bioactive molecules between cells, extracellular vesicles are being investigated as therapeutic agents for various diseases (See Figure 2). Their potential therapeutic [56] and drug delivery [57] applications are also being explored, offering new avenues for treating conditions such as cancer and inflammatory diseases. The capacity of extracellular vesicles to deliver therapeutic payloads directly to target cells offers a promising strategy for drug delivery. Furthermore, engineered extracellular vesicles can be designed to target specific cell types or tissues, enhancing therapeutic efficacy and reducing off-target effects. Extracellular vesicles can transport proteins, lipids, mRNA, and noncoding RNA, including microRNA.

Extracellular vesicles are involved in various diseases, including cancer, where they facilitate tumor growth, metastasis, and immune evasion [58]. Tumor-derived extracellular vesicles can promote angiogenesis, create a pre-metastatic niche, and suppress anti-tumor immune responses. Their role in cancer progression has made them attractive targets for therapeutic intervention. Dysregulation of extracellular vesicle-mediated communication has been implicated in the pathogenesis of neurological disorders. Extracellular vesicles also contribute to the spread of neurodegenerative diseases by facilitating the transfer of misfolded proteins, such as amyloid-beta and tau, between cells [59].

The surge in interest surrounding extracellular vesicles is reflected in the exponential increase in extracellular vesicle-related clinical trials, which aim to harness their diagnostic and therapeutic potential across a spectrum of diseases. In their systematic review, Mizenko et al. [60] examine the current landscape of extracellular vesicle-related clinical trials to elucidate key trends in clinical applications and methodological approaches, including the consideration of extracellular vesicle subpopulations. By analyzing data from publicly available clinical trial registries (e.g., clinicaltrials.gov), the authors identified 471 extracellular vesicle-related clinical trials encompassing indications across more than 200 diseases [60]. Diagnostic and companion diagnostic applications comprised most of these studies, with cancer being the most frequently targeted indication. Additionally, the ability of extracellular vesicles to cross biological barriers, including the blood-brain barrier, makes them particularly attractive for delivering therapeutics to the central nervous system.

Despite their promising potential, the use of extracellular vesicles in the clinical setting is restricted due to the lack of standardization in isolation and analysis methods [61]. Extracellular vesicles present both opportunities and challenges in clinical translation. The development of robust and scalable manufacturing processes is crucial for producing extracellular vesicle-based therapeutics.

In addition, extracellular vesicles hold promise as biomarkers for various diseases, including cancer, cardiovascular diseases, and neurological disorders. Extracellular vesicle-based diagnostics primarily aim to identify the physicochemical signatures of extracellular vesicles produced during the initiation and progression of disease [62]. Compared to conventional invasive tissue biopsies, tissue-derived extracellular vesicles can be found in peripheral circulation and isolated through minimally invasive sampling methods such as blood, urine, or saliva.

Among disease types, cancer-associated extracellular vesicles have been the most extensively studied. Dysregulation in extracellular vesicle cargo, encompassing proteins, nucleic acids, lipids, and even overall concentration, has been implicated in various disease states, underscoring their potential as diagnostic biomarkers and therapeutic targets [63]. Distinct extracellular vesicle signatures have also been observed in a range of other diseases, including neurodegenerative [64], cardiovascular [65], and infectious diseases [66]. The potential of extracellular vesicles extends beyond diagnostics, encompassing the monitoring of disease progression and treatment response [67]. This is particularly valuable for diseases that are challenging to diagnose, such as Alzheimer’s disease.

Exhibiting characteristics of drug delivery systems, biologics, and cell-based therapies, extracellular vesicles can function as nature’s own delivery vehicles, possess intrinsic biotherapeutic properties, and replicate the therapeutic effects of cellular therapies without the associated risk of uncontrolled proliferation [56].

**Figure 2 ijms-26-05896-f002:**
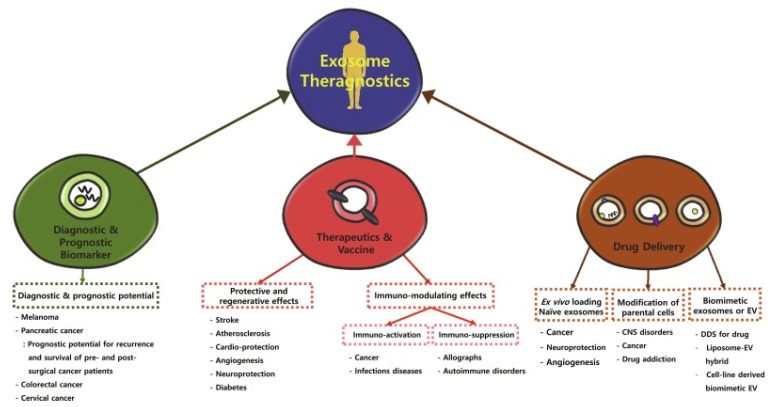
Summary of therapeutic and diagnostic (theragnostic) potential of exosomes (from Kang et al. [68]).

## 8. Resurrection of Darwin’s Pangenesis Theory

Despite Galton’s 1871 experiments showing no evidence for pangenesis, Darwin never wavered in his belief that all cells of the body “throw off” minute particles that may be transmitted to other parts of the body [69]. For a whole decade before he died in 1882, he engaged with a young physiologist, George Romanes, to carry out grafting experiments in plants in his attempts to prove his case. Much of that work was subsequently referred to by Romanes in his 3-volume 1893 publication, *Darwin and After Darwin* [8].

But no further experiments on Darwin’s theory were carried out until the 1950s, when a number of Russian scientists performed transfusion experiments on poultry and rabbits, showing evidence that the idea was correct. Liu has analyzed more than 50 papers published on these experiments and concludes that 45 gave positive results (see Liu et al., 2008 [11], for references and further explanation). Even before the proposal that EVs might be the missing process in pangenesis, Liu concluded that “Darwin’s Pangenesis contains a great truth and needs to be reconsidered” [11] (p. 149).

Since Liu’s careful historical work, the evidence that transmission of nucleotides, proteins, and metabolites can pass from the soma to the germline has become very strong. An article entitled “Bubbling Beyond the Barrier”, co-authored by one of us [18], has around 150 references to experiments showing how widespread the phenomenon is. Their conclusion is that “the idea that so many cases of GEI are expected to change an animal’s evolutionary fitness is re-igniting the historical debate between Lamarckian and Neo-Darwinian evolution” [18] (p. 10).

Time and again, advances in experimental methods enabling scientists to visualize objects and processes previously beyond the resolution of previous methods show that “progress is often achieved by a ‘criticism from the past’. Theories are abandoned and superseded by more fashionable accounts long before they have had an opportunity to show their virtues” [70] (p. 59).

## 9. Conclusions

The discovery of exosomes and other forms of extracellular vesicles transforms the theory of meridians in the various forms of Asian traditional medicine. There is no longer a conflict between Western and Eastern interpretations of the anatomy of the body. Furthermore, the specificity assumed in traditional medicine could, in consequence, be susceptible to standard scientific interpretations of the causality involved.

The same comment from Feyerabend in the previous section applies to the meridians of Asian Traditional Medicine. With the discovery of the targeted functionality of exosomes and other extracellular vesicles, a complete system of ancient medical treatment comes within reach of the methods of modern medical science.

Concerning the wired pathway, which the BongHan System claimed, it does not need to be fully identified as an additional circulatory system. The question is rather how versatile our living systems are. The extracellular vesicle pathway works as a general signaling mechanism, but there is nothing to prevent that system from using standard anatomical vessels. The specificity need not lie in the pathway but rather in what is transported, the targeted vesicle.

We conclude that vesicular communication can not only provide a process for the long-postulated meridians but also an explanation for the specificity of the targeting, which may arise naturally during embryonic development. The meridians may then follow the pathways of embryonic cell migration.

## Figures and Tables

**Figure 1 ijms-26-05896-f001:**
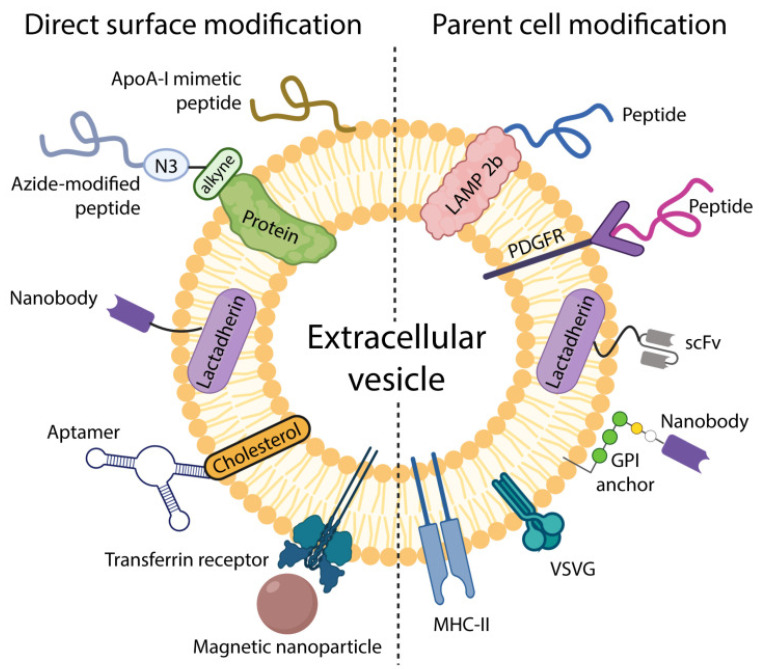
Schematic showing two strategies of EV surface modification. Left: Methods and moieties used via direct in vitro modification. Right: Surface modification via parental cell modification (from Frolova et al. [55]).

**Table 1 ijms-26-05896-t001:** Timeline of theories and experiments on hidden micro-level processes.

Date	Proposal, Theory, or Experiment	Positive or Negative
	NINETEENTH CENTURY	
1809	LAMARCK *Philosophie Zoologique* Gradual transformation of species by inheritance of acquired characteristics and first detailed Tree of Life	positive
1837	DARWIN Tree of Life sketch in Notebook B	neutral
1858	DARWIN & WALLACE Joint presentation of theory of Natural Selection	neutral
1859	DARWIN *The Origin of Species*	positive
1866	WALLACE proposes Natural Selection as sole process of evolution	negative
1868	DARWIN proposes existence of gemmules in *The Variation of Animals and Plants under Domestication*	positive
1871	GALTON performs transfusion experiments	negative
1883	WEISMANN proposes soma-germline BARRIER	negative
	TWENTIETH CENTURY	negative
1942	JULIAN HUXLEY adopts the Weismann Barrier in *Evolution: The Modern Synthesis*	
1950s	Around 50 studies on possible transfusion of micro-level processes by Russian scientists 45 out of 50 support the idea [11]	positive
1956	CRICK formulates Central Dogma of Molecular Biology	negative
1963	HUXLEY dethrones proteins in favour of DNA in 2nd edition of *Evolution: The Modern Synthesis*	negative
1967	Electron microscopy detects external vesicles thought to be cell debris [19]	neutral
	TWENTY-FIRST CENTURY	
2005	Smith & Spadafora show transfer of RNAs to germline cells	positive
2019	Noble proposes that external vesicles correspond to Darwin’s gemmules	positive
2024	Phillips & Noble (2024) [18] summarize more than 100 experiments showing evidence of “bubbling beyond the barrier”	positive

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
