# Peer review of "Extracellular Vesicles as Targeted Communicators in Complementary Medical Treatments"

_ijms, 2025, doi:10.3390/ijms26125896_

Round 1

Reviewer 1 Report

Comments and Suggestions for Authors

In this manuscript, Keehyun Earm and colleagues describe an attractive theory that propagation through EVs and exosomes may form the communication pathways commonly described as meridians in Chinese medical treatments, including massage and acupuncture. To adhere to the title of this article, “Extracellular Vesicles as Targeted Communicators in Complementary Medical Treatments,” the authors should describe more about extracellular vesicles as targeted communicators in drug delivery in depth. In addition, the authors could perhaps summarize the molecular composition, targeting mechanism, and advantages and disadvantages of vesicles as targeted communicators with more figures and tables.

  • It is suggested to reorganize the parts including 2. Fields and hidden micro-level processes”, “3. Vesicles and exosomes as Darwin’s gemmules” and “8. Resurrection of Darwin’s pangenesis theory” with a summarized clear history timeline to better reveal the development of the subject.
  • The molecular composition of extracellular vesicles should be described in detail in “6. Molecular Composition and Targeting Mechanisms of Extracellular Vesicles”.
  • In “7. Therapeutic potential and use as markers of disease states”, perhaps a graphical summary of the therapeutic potential of vesicles and the diseases for which they can be used as markers.
  • References in this article are not formatted properly, it is recommended to standardize the reference format according to the journal requirements and make sure that the cited references are closely related to the content of the article.

Reviewer 2 Report

Comments and Suggestions for Authors

While the authors provide a broad, well referenced review of extracellular vesicles (EVs) and provide and interesting tie between EVs and Darwin’s gemmules there are a number of significant issues with the manuscript.

The authors postulate the interesting hypothesis that EVs contribute to the

meridians utilized in traditional therapies such as massage and acupuncture and this therapeutic effect is based upon the preservation of EV release and uptake mechanisms shared by tissues that are in proximity during development.  This would suggest that EVs during development are involved in regulation of proximal tissue and that the mechanism of this regulation is preserved as the tissues mature and change relative positions during tissue development.  Thus, one would anticipate that there would be matching EV protein/ tissue protein interactions (i.e. receptor ligand interactions) that match their model of initial proximity and subsequent spatial separation.  While the authors give a number of examples of EV acting on distant tissues as well as provide a number of examples of targeting peptides/proteins and their receptors, they do not demonstrate any evidence that any of these known interactions (or even proximity of these players) occur in adjacent tissues during development. Nor do they present evidence that these known interactors (i.e. receptor/ligand pairs) are present selectively on a given body surface/organ meridian. The later would support the idea that such a meridian specific EV/target tissue interaction drives the therapeutic effect.

The acupuncture literature has examples of specific EV miRNA produced from specific acupuncture sites and targeting specific cell types (acupuncture induce EV production which in turn effects on neural progenitors, vasculogenic and immune functions mediated by specific miRNAs).  As with the receptor/ligand discussion above, it may be possible to examine spatial localization of the cells that produce these miRNAs with the EV/miRNA target cells during development and adulthood looking for early proximity.    

Thus overall, while the hypothesis is interesting there was no attempt to bring in supportive data for the model or even utilize existing data to support a testable model.

The overall structure of the manuscript was highly problematic as the EVs/Darwin’s gemmules discussion as written was unrelated to the meridian hypothesis. These seem like components of separate papers.

The focus on therapeutic cell specific targeting of EVs supported the idea that EVs could utilize cell specific targeting mechanisms but was by definition artifactual and thus was not as useful as discussing intrinsic EV cell type targeting mechanisms.

Round 2

Reviewer 1 Report

Comments and Suggestions for Authors

Agree to acceptance

Reviewer 2 Report

Comments and Suggestions for Authors

The authors addressed many of my concerns.  I feel that the manuscript is acceptable for publication.